# Anion-Responsive Fluorescent Supramolecular Gels

**DOI:** 10.3390/molecules27041257

**Published:** 2022-02-13

**Authors:** Giacomo Picci, Matthew T. Mulvee, Claudia Caltagirone, Vito Lippolis, Antonio Frontera, Rosa M. Gomila, Jonathan W. Steed

**Affiliations:** 1Dipartimento di Scienze Chimiche e Geologiche SS 554 Bivio per Sestu, Università degli Studi di Cagliari, 09042 Monserrato, CA, Italy; gpicci@unica.it (G.P.); lippolis@unica.it (V.L.); 2Department of Chemistry, University of Durham, South Road, Durham DH1 3LE, UK; mattmulvee@gmail.com; 3Departament de Química, Universitat de les Illes Balears, Ctra. de Valldemossa km 7.5, 07122 Palma de Mallorca, Baleares, Spain; rosa.gomila@uib.es

**Keywords:** supramolecular chemistry, anion recognition, stimuli-responsive gels, noncovalent interactions, DFT calculations

## Abstract

Three novel bis-urea fluorescent low-molecular-weight gelators (LMWGs) based on the tetraethyl diphenylmethane spacer—namely, **L1**, **L2**, and **L3**, bearing indole, dansyl, and quinoline units as fluorogenic fragments, respectively, are able to form gel in different solvents. **L2** and **L3** gel in apolar solvents such as chlorobenzene and nitrobenzene. Gelator **L1** is able to gel in the polar solvent mixture DMSO/H_2_O (H_2_O 15% *v*/*v*). This allowed the study of gel formation in the presence of anions as a third component. An interesting anion-dependent gel formation was observed with fluoride and benzoate inhibiting the gelation process and H_2_PO_4_^−^, thus causing a delay of 24 h in the gel formation. The interaction of **L1** with the anions in solution was clarified by ^1^H-NMR titrations and the differences in the cooperativity of the two types of NH H-bond donor groups (one indole NH and two urea NHs) on **L1** when binding BzO^−^ or H_2_PO_4_^−^ were taken into account to explain the inhibition of the gelation in the presence of BzO^−^. DFT calculations corroborate this hypothesis and, more importantly, demonstrate considering a trimeric model of the **L1** gel that BzO^−^ favours its disruption into monomers inhibiting the gel formation.

## 1. Introduction

Supramolecular chemistry is an area of chemistry of great interest that aims to investigate novel multimolecular systems in which the components are held together reversibly by noncovalent interactions. Indeed, although the presence of forces defined ‘weak’, the self-assembly of particular molecular building blocks produces functional architectures strong enough to bind specific guest species, ensuring, at the same time, a certain degree of flexibility to enable the supramolecular systems to withstand small perturbations such as external stimuli and to maintain the self-assembled structure. Starting from the fundamental studies of host-guest chemistry described for the first time by Jean M. Lehn [1], the interest in new supramolecular systems have found applications in many different fields, such as in cancer nanotheranostics [2], medicinal chemistry, sensing, extraction [3] and development of novel materials which are ready to move into the market or are already commercially available [4,5]. 

In particular, supramolecular gels are an appealing and relatively new class of materials that have recently attracted the attention of the scientific community for their peculiar physical and chemical properties [6,7,8,9]. While classic gels are mostly based on polymer gelators, supramolecular gels are based on low-molecular-weight gelators (LMWGs), small molecules able to entrap solvent molecules into a nanostructured 3D network when self-assembling via noncovalent interactions such as hydrogen bonds, Van der Waals interactions, π-π stacking, and also metal coordination [10,11,12]. One of the most interesting features of supramolecular gels is the possibility of a stimuli-responsive behaviour which is a direct consequence of the reversibility of the noncovalent interactions at the basis of the 3D network formation. Indeed, a change of temperature, pH, or the presence of metal ions or anions can cause a gel-sol transition or change the properties of the material itself improving, for example, its stiffness or its self-healing properties [13,14,15,16,17]. The possibility to form H bonds between anions and ureas or amides has been exploited by supramolecular chemists for the design of anion-selective receptors or chemosensors [18]. Urea is also a very common motif for the development of LMWGs due to the possibility of forming intermolecular H bonds between urea molecules. For these reasons, anion-responsive gels based on ureas are commonly found in the literature [11,19]. Despite the fact that this class of compounds is quite popular, some challenges still need to be addressed, such as the control of the properties of the materials in the presence of different anions and the selective response of the material to the presence of one specific anion. 

Bis-urea gelators featuring a hydrophobic diphenylmethane derived central are very efficient for the formation of robust gels and metallogels [20,21,22,23,24,25].

In this study, we describe the use of this platform in the construction of anion-tuned, fluorescent gels based on our previous research on the design of fluorescent materials [26,27,28,29,30,31]. We report here three novel gelators (**L1**–**L3**, Figure 1), their self-assembling properties in the formation of gels, and their anion-responsive and photophysical behaviour. Dansyl and quinoline (in **L2** and **L3**, respectively) fluorophores were chosen, as they could act both as H bond donors (sulphonamide moiety in dansyl) and acceptors (nitrogen in quinoline). The indole moiety in **L1** was incorporated because indole-based receptors exhibit excellent selectivity in anion binding, sensing, and transport [32,33,34,35,36,37,38,39,40,41,42,43]. Indeed, the indole NH moiety exhibits striking H-bond donor properties and can offer the possibility to reinforce the host-guest interaction via cooperative H bonds if other H-bond donor NH groups (such as urea or amide) are present in the molecular skeleton. Surprisingly, to the best of our knowledge, only two examples of supramolecular gels have been reported previously in which the indole moiety plays a role in the self-assembling process [44,45]. 

## 2. Results

### 2.1. Synthesis and Characterisation of Gels

Gelators **L1**–**L3** were synthesised by reacting the bis-isocyanate bis(3,5-diethyl-4-isocyanatophenyl)methane [20] with the appropriate amine following the synthetic procedures reported in the Experimental Section (see Appendix A for ^1^H, ^13^C of **L1**, **L2**, and **L3**, and ESI mass spectra of **L2** and **L3**). 

Gelation tests were carried out at different concentrations in a wide range of solvents (nitrobenzene, chlorobenzene, nitromethane, 1,4-dioxane, tetrahydrofuran, acetonitrile, dichloromethane, chloroform, methanol, ethanol, 1-isopropanol, 2-isopropanol, 1-butanone, ethyl acetate, dimethyl sulphoxide (DMSO), DMSO/water in various ratios). Samples were sonicated, then heated gently to dissolve the solid, and subsequently allowed to cool to room temperature. Gelators **L1**–**L3** showed low solubility in most of the solvents used. Gels formed with apolar solvents such as nitrobenzene and chlorobenzene for **L2** and **L3**, while, in these solvents, **L1** was insoluble (Appendix A). The only solvent, among those considered, able to solubilise **L1** was DMSO. Hence, we decided to conduct gelation tests in a mixture of DMSO and water (as the antisolvent) at different percentages of water. Under these conditions, we were able to obtain a gel for **L1** at 2% *w*/*v* in DMSO/H_2_O (H_2_O 15% *v*/*v*). These conditions were found to be optimal, with 2% *w*/*v* being the critical gelation concentration; on increasing the amount of water up to 20% in the solvent mixture, partial precipitation of the gelator was observed. In the case of **L2** and **L3**, when water was added to the DMSO solution, a precipitate was always observed. The gels formed by **L2** and **L3** in nitrobenzene and chlorobenzene were semitransparent, as shown in Figure 1. Gels were dried in a vacuum desiccator for 24 h, to yield xerogels which were then coated with platinum and imaged by SEM. SEM images showed an entangled network of thick fibres for xerogels of **L2** and **L3** (Appendix A). Similar fibres were observed also in the gels previously reported bearing the central diphenyl methane spacer [23]. 

Oscillatory stress sweep rheometry measurements were conducted to probe the viscoelastic properties of gels of **L2** and **L3** in both nitrobenzene and chlorobenzene. As shown in Figure 1, the solid-like properties of the materials were suggested by the ratio between the *G′* (storage modulus) and the *G″* (loss modulus) with *G′* higher than *G″* of around an order of magnitude. The weak nature of the gels obtained in the two solvents for both **L2** and **L3** was suggested by the low values of *G′* between 7000 Pa and 5000 Pa. Nonetheless, the yield stress around 100 Pa demonstrated the stability of the materials at low yield stress. In particular, for both **L2** and **L3**, the gels with better solid-like properties were obtained in nitrobenzene (Figure 1a,c for **L2** and **L3**, respectively). Although it is not possible to compare the gels’ properties in nitrobenzene, as they were formed at different gelator’s concentrations (1% *w*/*v* vs. 0.75% *w*/*v* for **L2** and **L3**, respectively), in chlorobenzene, the gel formed by **L2** exhibits a higher elastic modulus than **L3** at the concentration of 0.75% *w*/*v* (*G′* = 1800 and *G′* = 400 for **L2** and **L3**, respectively (Figure 1b,d). 

The most interesting results were obtained for the gel of **L1** at 2% *w*/*v* in DMSO/H_2_O (H_2_O 15% *v*/*v*). In this case, as shown in Figure 2a, the strong solid-like properties of the material were highlighted by the high value of *G′*, the difference between *G′* and *G″* significantly higher than an order of magnitude, and the high value of yield stress of about 1000 Pa. The material also exhibits weak strain overshoot [46]. The SEM image of the corresponding xerogel (Figure 2b) showed a dense network of fibres. 

### 2.2. ^1^H-NMR Spectroscopic Studies

Host-guest interactions within supramolecular gels can modulate the self-association properties of the gelators [47]. Herein, we investigated the gelation process in the presence of an anion guest as a third component. The most suitable system to study the anion–gelator host-guest interactions is **L1**, as it forms a gel in a polar medium (i.e., DMSO/H_2_O (H_2_O 15% *v*/*v*). Gelation tests on **L1** in the presence of different anions (Cl^−^, BzO^−^, H_2_PO_4_^−^, F^–^ as tetrabutylammonium salts) in a 1:2 ratio with respect to the gelator were performed in DMSO/H_2_O (H_2_O 15% *v*/*v*) at 2% *w*/*v* of **L1**. The procedure of the preparation of gels is described in SI. The inversion tube tests of the samples (Figure 3) showed the different responses of the gelation process in the presence of anions. 

Upon the addition of the different aqueous anion solutions to the DMSO solution of **L1**, various behaviours were observed. In particular, the presence of chloride (Figure 3a) did not affect the gelation process of **L1**. The addition of fluoride (Figure 3c), instead, completely inhibited the gel formation. In the case of benzoate, the addition of the anion inhibited the gelation process as well, although a certain aggregation tendency was detected, as shown in Figure 3d. The behaviour observed in the presence of the dihydrogen phosphate was particularly interesting. Indeed, the formation of the gel did not occur immediately as in the presence of chloride, but it was delayed, occurring in the following 24 h after the mixing of the components (Figure 3b). After that time, no further changes were observed.

To understand the interactions between **L1** and the anion guests in solution, anion-binding studies were conducted by means of ^1^H-NMR titrations using DMSO-*d*_6_/0.5% water as a solvent. The results obtained by ^1^H-NMR titrations provide insight into the interactions that could occur between the anions and the gelator during the gelation process and rationalise the behaviour of the system.

Stability constants from the ^1^H-NMR titration curves were calculated by fitting the data to a 1:2 (receptor/anion molar ratio) binding model by using Bindfit [48,49], as reported in Table 1. Titration data could not be fitted by a 1:1 model. 

In the case of fluoride, as expected, the deprotonation of the urea NH protons of **L1** was observed in solution which would destroy the H-bond network involved in the self-assembling process that is the basis of the gel formation. In contrast, in the case of chloride, the interaction between **L1** and the anion in solution is extremely weak (Appendix A); therefore, the gelation is essentially unaffected by chloride, and gel formation was observed. The gelator binds strongly to H_2_PO_4_^−^, particularly BzO^−^, and for both anions, the indole NH and the urea NH protons of the gelator undergo a significant downfield shift, implying significant binding cooperativity between the two types of NH hydrogen bond donors (Appendix A). Moreover, the possibility of forming stacking interactions between the anions (see below) that stabilise the adduct should be taken into account in the case of benzoate. In the case of H_2_PO_4_^−^, on the other hand, although the interaction between **L1** and the anion is strong, the anion association may be more labile and weaker than in the case of benzoate, leading to the formation of the gel 24 h after mixing. 

### 2.3. Theoretical Calculations

DFT calculations in DMSO (BP86-D3-COSMO/def2-TZVP level of theory; see Theoretical Methods for details) were performed in order to investigate the geometry of the 1:2 **L1** complexes with BzO^−^ and H_2_PO_4_^−^ anionic guests (see Appendix A for atomic coordinates). Moreover, the relative ability of the latter to interact with the host was studied energetically. The results are shown in Figure 4, which reveals relatively different binding modes for BzO^−^ and H_2_PO_4_^−^ anions. In the case of H_2_PO_4_^−^, there is no interaction between the two guests, and each anion establishes a bifurcated acceptor O···HN(urea) H bond with the urea moiety of **L1** and a short O···H-N H bond with the indole ring. In the case of BzO^−^ guest, both guest units interact with each other establishing a T-shape edge-to-face contact (see Appendix A for further details and distances). Moreover, each BzO^−^ anion establishes three H-bonding contacts similar to those described above for H_2_PO_4_^−^ anion. The main difference is that for BzO^−^, the three H-bonding distances are shorter (ranging from 1.75 to 1.83 Å), offering evidence of strong interactions. In contrast, for the H_2_PO_4_^−^ anion, one of the N-H···O H-bonds with the urea moiety is longer than 2 Å, thus revealing a weaker H-bond interaction. This is likely due to the difference in the O-Y-O angle (Y = P or C), tetrahedral for Y = P, and trigonal planar for Y = C, allowing a better complementarity in the case of BzO^−^. Moreover, BzO^−^ is a better H-bond acceptor than H_2_PO_4_^−^. Therefore, the calculated geometric features of both complexes provide evidence of a stronger H-bond network in the BzO^−^ complex; this is accompanied by the extra stabilisation in **L1**·(BzO^−^)_2_ due to the guest-guest interaction. The stronger interaction of BzO^−^ with **L1**, compared with H_2_PO_4_^−^, is further supported by the significant negative value of −14 kcal mol^−1^, calculated for the free energy change associated with the substitution reaction **L1**·(H_2_PO_4_^−^)_2_ + 2 BzO^−^ → **L1**·(BzO^−^)_2_ + H_2_PO_4_^−^ in DMSO solution. 

In addition, we used a trimeric assembly of **L1** as a model of the supramolecular polymer that is likely formed in the initial steps of the gelation process (see Appendix A for atomic coordinates). The optimised trimer is shown in Figure 5, presenting a strong complementarity of the monomers. The central monomer in the trimer establishes 12 strong NH···OC bonds as donor and acceptor via the urea and indole moieties. Moreover, four sets of parallel-displaced π-stacking interactions are also formed involving the indole and the biphenyl moieties. Such a combination of interactions explains the strong gelation ability of **L1**. Interestingly, we evaluated the ability of BzO^−^ and H_2_PO_4_^−^ anions to break the trimer into monomers to explain the different behaviour observed experimentally (i.e., BzO^−^ preventing gelation). As a satisfactory finding, the transformation of the trimer into three monomers (interacting with the anions) is favoured for benzoate anion and disfavoured for phosphate (Figure 5, bottom), thus providing an explanation for the inhibition of gelation of **L1** in the presence of benzoate. 

We also analysed the formation of the trimers for ligands **L2** and **L3** as a possible initial mechanism for the gel formation. The optimised geometries are given in Figure 6, showing a similar self-assembly mechanism, compared with **L1**, which is a combination of H bonds and π-stacking interactions. In the case of **L2**, eight H bonds (four as donors and four as acceptors) interconnect the central monomer to the other two by means of the urea units. Moreover, three groups of π-stacking interactions further stabilise the trimeric assembly involving the inner (in pink) and outer (in green) aromatic rings. In the case of gelator **L3**, the central monomer establishes 12 H bonds, similar to **L1**, due to the extra participation of the sulphonamide NH group (Figure 6, right). The NH···OC H-bonding distances range from 1.86 Å to 2.24 Å, and the π-stacking distances from 3.24 Å to 3.43 Å, similar to those found for the trimer of **L1**.

### 2.4. Fluorescence Studies

The three gelators **L1**–**L3** were all designed to contain fluorogenic fragments in anticipation of gelation- and anion-induced changes to their emission profile. We investigated the fluorescence properties of **L1**–**L3** in solution in DMSO and in their gels. The choice of the solvent depended on the low solubility of the compounds. 

In the solutions, absorption (Appendix A) and emission properties of **L1**–**L3** resemble those of the free fluorophores (λ_em_ = 322 nm, 520 nm, 445 nm for **L1**, **L2**, and **L3**, respectively (Figure 6 and Appendix A). Emission spectra of the gels for **L1** (DMSO/H_2_O, with H_2_O 15% *v*/*v*), **L2**, and **L3** (both in chlorobenzene) were also recorded. The **L1** gel has a low-intensity emission band centred at 402 nm (Figure 7a), while the emission of **L2** gel is centred at 520 nm (Figure 7b). In the case of **L3**, the gel is not fluorescent.

The shift in the maximum of the emission observed in solution and in the gel phase, for both **L1** and **L2**, could be attributed to the dependence of the photophysical properties of the fluorophores on the solvent polarity and increased aggregation [50,51]. Indeed, the hypsochromic shift of 30 nm observed moving from solution to gel phase for **L2** could be attributed to the different solvents used (DMSO and chlorobenzene, for solution and gel, respectively). The shift, which, in this case, is bathochromic, is even more marked in the case of **L1**, in which the presence of 15% water used to form the gel could also explain the lower intensity of the emission, compared with the solution state. Additionally, in the case of **L2**, the gel is less fluorescent than the solution.

Finally, to further understand the behaviour of **L1** in the presence of BzO^−^ or H_2_PO_4_, we investigated the emission properties of the gelator with these two anion guests (as tetrabutylammonium salts) in DMSO solution. 

The addition of increasing amounts of H_2_PO_4_^−^ caused an enhancement in the fluorescence characteristic of the free gelator (Appendix A). In contrast, an almost negligible effect (weak quenching) was observed in the presence of BzO^−^ (Appendix A). Similar behaviour with H_2_PO_4_^−^ and BzO^−^ was observed by the Beer group with indolocarbazole receptors [52]. Fluorescence titrations confirmed the formation of 1:2 **L1**/anion adducts (see Appendix A for H_2_PO_4_^−^ and BzO^−^, respectively).

## 3. Materials and Methods

All reactions were performed in oven-dried glassware under a slight positive pressure of nitrogen. ^1^H-NMR (600 MHz) and ^13^C NMR (126 MHz) spectra were determined on a 600 MHz Bruker and on 300 MHz Bruker. Chemical shifts for ^1^H NMR were reported in parts per million (ppm), calibrated to the residual solvent peak set, with coupling constants reported in Hertz (Hz). The following abbreviations were used for spin multiplicity: s = singlet, d = doublet, t = triplet, q = quadruplet, m = multiplet. Chemical shifts for ^13^C NMR spectra were reported in ppm, relative to the central line of a septet at δ = 39.52 ppm for deuteriodimethyl sulphoxide. Infrared All solvents and starting materials were purchased from commercial sources where available (Merck Europe, Fluorochem UK). Proton NMR titrations were performed by adding aliquots of a putative anionic guest (such as the TBA salt, 0.075 M) in a solution of the receptor (0.005 M) in DMSO-*d*_6_/0.5% water.

FTIR spectra of all solids were taken in the lab on a PerkinElmer Spectrum 100 Series spectrometer. Oscillatory stress sweep experiments were performed between 0.1 Pa and 1000 Pa, at a constant frequency of 1 Hz, on a TA instrument AR 2000 rheometer equipped with a rough plate geometry. When preparing the sample, 2 mL of gelator solution was transferred to a sealed glass cylinder on the lower plate. The gels were allowed 30 min to equilibrate before the geometry was lowered onto the sample at a predetermined gap of 2.2–2.5 mm, and the glass cylinder was gently removed before running the experiment. 

### 3.1. 1,1′-(methylenebis(2,6-diethyl-4,1-phenylene))bis(3-(1H-indol-7-yl)urea) (L1)

To a stirred solution of 7-aminoindole (0.18 g, 1.38 mmol) in CH_2_Cl_2_ (10 mL), a solution of bis(3,5-diethyl-4-isocyanatophenyl)methane (0.25 g, 0.69 mmol) in CH_2_Cl_2_ (5 mL) was added dropwise, under a N_2_ atmosphere. The solution was kept stirring at room temperature for 24 h. When the reaction was stopped, a precipitate was observed and isolated by filtration. The solid was further washed with CH_2_Cl_2_ (3 × 5 mL) and dried under reduced pressure, to remove the residual solvent obtaining the product as a grey solid. Y = 85%, 0.37 g; ^1^H-NMR (600 MHz, 298 K), δ_H_: 10.68 (s, 1H), 8.68 (broad, 1H), 7.68 (s, 1H), 7.30 (t, 1H, J =3.0 Hz), 7.23 (d, 1H, J = 8.0 Hz), 7.11(d, 1H, J = 8.0 Hz), 7.03(s, 2H), 6.91 (t, 1H, J = 8.0 Hz), 6.40 (s, 1H), 3.87(s, 1H), 2.60 (q, 4H, 8.0 Hz), 1.15 (t, 6H, J = 6.0 Hz);^13^C-NMR (126 MHz, 298 K), δ_C_:14.69, 24.47, 40.87, 101.56, 114.78, 119.13, 124.77, 125.00, 126.20, 129.34, 132.01, 139.80, 141.65, 154.49. Elemental analysis calcd % (found%) for C_39_H_42_N_6_O_2_: C% 74.73 (74.59), H% 6.75 (6.81), N% 13.41 (13.39). 

### 3.2. N,N′-(((((methylenebis(2,6-diethyl-4,1-phenylene))bis(azanediyl))bis(carbonyl))bis(azanediyl)bis(ethane-2,1-diyl))bis(5-(dimethylamino)naphthalene-1-sulfonamide (L2)

To a stirred solution of N-(2-aminoethyl)-5-(dimethylamino)naphthalene-1-sulphonamide (0.32 g, 1.10 mmol) in CH_2_Cl_2_ (5 mL), a solution of bis(3,5-diethyl-4-isocyanatophenyl)methane (0.20 g, 0.55 mmol) in CH_2_Cl_2_ (5 mL) was added dropwise, under a N_2_ atmosphere. The solution was kept stirring at room temperature for 24 h. When the reaction was stopped, a precipitate was observed and isolated by filtration. The solid was further washed with CH_2_Cl_2_ (3 × 5 mL) and dried under reduced pressure, to remove the residual solvent obtaining the product as a yellow solid. Y = 81% 0.42 g; ^1^H-NMR (600 MHz, 298 K), δ_H_: 8.46 (d, 1H, J = 6.0 Hz), 8.26 (d, 1H, J = 6.0 Hz), 8.09 (d, 1H, J = 6.0 Hz), 7.94 (broad, 1H), 7.63-7.56 (m, 2H), 7.39 (s, 1H), 7.25 (d, 1H, J = 6.0 Hz), 6.92 (s, 2H), 6.06 (broad, 1H) 3.78 (s, 1H), 3.03 (broad, 2H), 2.82(s, 6H), 2.76 (broad, 2H), 2.40 (q, 4H, J = 6.0 Hz), 1.00 (t, 6H, 12.0 Hz); ^13^C-NMR (126 MHz, 298 K), δ_C_: 14.61, 24.32, 40.78, 43.10, 44.95, 115.10, 119,15, 123.54, 126,21, 127,82, 128.25, 129.07, 129.39, 135.74, 142.00, 151.35, 156.83. (ESI^+^): *m*/*z*: exp: 949.7273 [M]^+^ calc: 949.4469 [M]^+^ Elemental analysis calcd % (found%) for C_51_H_64_N_8_O_6_S_2_: C% 64.53 (64.48), H% 6.80 (6.77), N% 11.80 (11.79). 

### 3.3. 1,1′-(methylenebis(2,6-diethyl-4,1-phenylene))bis(3-(quinolin-8-yl)urea) (L3)

To a stirred solution of 8-aminoquinoline (0.16 g, 1.10 mmol) and triethylamine (6.60 mmol) in CH_3_CN, a solution of bis(3,5-diethyl-4-isocyanatophenyl)methane (0.20 g, 0.55 mmol) in CH_3_CN (5 mL) was added dropwise, under a N_2_ atmosphere. The solution was heated at 70 °C and stirred for 24 h. When the reaction was cooled, a precipitate was observed and isolated by filtration. The solid was further washed with dichloromethane (3 × 5 mL) and dried under reduced pressure, to remove the residual solvent obtaining the product as a white solid. Y = 41%, 0.15 g; ^1^H-NMR (600 MHz, 298 K), δ_H_: 9.75 (s, 1H) 8.93 (s, 1H), 8.77 (s, 1H), 8.52 (d, 1H, J = 6.0 Hz), 8.38 (d, 1H, J = 6.0 Hz), 7.62 (broad, 2H), 7.50 (s, 2H), 7.03 (s, 1H), 3.89 (s, 1H), 2.65(q, 4H, J = 6.0 Hz), 1.34 (t, 6H, J = 6.0 Hz); ^13^C-NMR (126 MHz, 298 K) 14.77, 24.55, 40.76, 113.99, 119.19, 121.89, 126.33, 127,20, 127,91, 131.88, 136.53, 137.60, 139.60, 141.87, 148.12, 153.84; (ESI^+^): *m*/*z*: exp: 651.3467 [M]^+^ calc: 651.3447 [M]^+^; Elemental analysis calcd % (found%) for C_41_H_42_N_6_O_2_: C% 75.67 (75.69), H% 6.50 (6.51), N% 12.91 (12.89). 

### 3.4. DFT Calculations 

All calculations were performed using the program Turbomole 7.0 [53]. The geometry optimisations were performed without imposing any constrain at the MARIJ-PB86-D3/def2-TZVP and COSMO-RS to take into account solvation effects [54]. This solvation method (RS = real solvent) is an improved version of the conductor-like solvation model (COSMO) [55]. This level of theory is a reasonable compromise between the accuracy of the results and the size of the system (276 atoms for the trimer) and uses the D3 [56] dispersion correction, which is convenient for interactions in which aromatic rings are involved.

## 4. Conclusions

We reported here three new fluorescent bis-urea LMWGs (**L1**–**L3**) bearing a diphenylmethane spacer and indole, dansyl, and quinoline substituents as fluorogenic fragments. While **L2** and **L3** form gels in apolar solvents such as chlorobenzene and nitrobenzene, **L1** gels are formed with the polar solvent mixture of DMSO/H_2_O (H_2_O 15% *v*/*v*) at 2% *w*/*v*. Gelator **L1** represents a rare example of a diphenylmethane bis-urea gelator able to gel in a quite polar solvent mixture [20,21].This allowed us to study the gel formation in the presence of anions as a third component. The weakly bound chloride does not inhibit the gelation process, while deprotonation by fluoride prevents it entirely. The relatively strongly bound H_2_PO_4_^−^ acts the delay gel formation but does not prevent it after 24 h, while the very strongly bound benzoate significantly hampers gelation. These observations were rationalised by studying the anion affinity of **L1** in solution DMSO-*d*_6_/0.5% water by means of ^1^H-NMR titration. In particular, the difference in the cooperativity of the two types of NH H-bond donor groups (one indole NH and two urea NHs) on **L1** when binding BzO^−^ or H_2_PO_4_^−^ in solution may explain the gel formation of the system **L1**/H_2_PO_4_^−^ in DMSO/H_2_O (H_2_O 15% *v*/*v*) and the inhibition of the gelation in the presence of BzO^−^. DFT calculations corroborate this hypothesis, indicating considerable stabilisation by benzoate, compared with H_2_PO_4_^−^. More importantly, the calculations with a trimeric model of **L1** gel provide evidence indicating that its disruption into monomers is not favoured in the presence of H_2_PO_4_^−^, whereas the contrary is found for BzO^−^. Overall, this research confirms the utility of the tetraethyldiphenyl methane spacer in promoting supramolecular gel formation and demonstrates that the introduction of the highly effective anion-binding indole group in the molecular skeleton of the gelator allows the control of gel formation in the presence of anions by modulating the fibre assembly process formation via the cooperation of indole NH and urea NHs. This might open new and interesting perspectives in the design of anion-responsive supramolecular gels with potential biomedical or environmental applications.

## Data Availability

Fittings of the ^1^H NMR titrations can be found at http://app.supramolecular.org/bindfit/view/a88ec1e2-c0bb-475e-9a63-a96b4437b7da (accessed on 30 January 2022), for TBABzO, http://app.supramolecular.org/bindfit/view/27cb44e8-1836-490b-8cf4-00bdb6f99620 (accessed on 30 January 2022), for TBACl, and http://app.supramolecular.org/bindfit/view/89fddba8-cce7-4e44-b0ac-050954774253 (accessed on 30 January 2022), for TBAH_2_PO_4_.

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
