# Peer review of "Anion-Responsive Fluorescent Supramolecular Gels"

_molecules, 2022, doi:10.3390/molecules27041257_

Round 1
Reviewer 1 Report
This manuscript describes a family of three novel bis-urea fluorescent LMWGs based on the tetraethyl diphenylmethane spacer were studied. And the effect of anions as the third component on gel formation was further studied. At the same time, the density function calculation is used to verify the experimental results. At the same time, 1H-NMR titrations and DFT calculations are used to research and verify the experimental results. This is a very innovative and scientifically valuable approach. This manuscript can be published in molecules after the following revisions.
- The gelation process in the presence of an anion guest as a third component should be described in more detail, and the action time of different anions should be increased, and the comparative pictures should be given.
- Please unify the format in the article, such as the annotation in Figure 3.
- The introduction is too brief, the summary of supramolecular development and its application should be increased. In order to support this statement, the following recently published important related papers should be cited: Chem. Soc. Rev. 2021, 50, 2839; J. Am. Chem. Soc. 2020, 142, 2228; Angew. Chem. Int. Ed. 2021, 60, 8115.
Author Response
We would like to thank the reviewer for evaluating the manuscript and for their nice comments.
- The gelation process in the presence of an anion guest as a third component should be described in more detail, and the action time of different anions should be increased, and the comparative pictures should be given.
As suggested by the reviewer we described in the Discussion session the gelation process in the presence of the anion guests and we added the experimental procedure in SI. As described in the manuscript the anion was dissolved in water and added to a DMSO solution of the gelator. The mixture was heated up and then allowed to cool. We observed the samples until there was no further changes. In the presence of chloride the gel formed immediately, in the presence of dihydrogen phosphate it took 24h to form. No further changes were observed after that time. The relevant pictures relative to the end of the gelation process are already present in Figure 3.
- Please unify the format in the article, such as the annotation in Figure 3.
The format was unified
- The introduction is too brief, the summary of supramolecular development and its application should be increased. In order to support this statement, the following recently published important related papers should be cited: Chem. Soc. 2021, 50, 2839; J. Am. Chem. Soc. 2020, 142, 2228; Angew. Chem. Int. Ed. 2021, 60, 8115.
- The introduction was improved and the suggested references added.
Reviewer 2 Report
This manuscript is described bis-urea low molecular weight gelators (LMWG). Interestingly, each of these LMWGs were gelled in different solvents and had different luminescence properties. The author fully investigated the interaction between LMWG and solvent and analyzed the mechanism of LMWG gelation. On the other hand, the analysis was only for L1 and the analysis of L2 and L3 was poor and insufficient. Thus, after the authors have addressed the comments below, this work would be suitable for publication in Molecules.
- Abbreviations should be used after defining the formal name. For example, “LMWG” in Abst line 1.
- The authors should show a titration plot with a receptor to anion ratio of 1: 2. (page 5, line 147).
- Compared to L1, there are few data showing the mechanism of gelation of L2 and L3. Data needs to be added to explain the mechanism of L2 and L3 gelation.
Author Response
- Abbreviations should be used after defining the formal name. For example, “LMWG” in Abst line 1.
This was corrected in the text.
- The authors should show a titration plot with a receptor to anion ratio of 1: 2. (page 5, line 147).
The required plots were added in the SI.
- Compared to L1, there are few data showing the mechanism of gelation of L2 and L3. Data needs to be added to explain the mechanism of L2 and L3 gelation.
As suggested by the referee we added an analysis of the formation of the trimers for ligands L2 and L3 as a possible initial mechanism for the gel formation. The optimized geometries were added in the manuscript in Figure 6.
Round 2
Reviewer 2 Report
The revisions made by the authors are adequate and I like to recommend publication as it is.